# Incorporation of Ag-ZnO Nanoparticles into PVDF Membrane Formulation to Enhance Dye Retention, Permeability, and Antibacterial Properties

**DOI:** 10.3390/polym17091269

**Published:** 2025-05-06

**Authors:** Baha Chamam, Roua Ben Dassi, Jraba Abderraouf, Jean Pierre Mericq, Catherine Faur, Ismail Trabelsi, Lassaad El Mir, Marc Heran

**Affiliations:** 1Laboratory of Treatment and Valorization of Water Rejects, Water Research and Technologies Center, Borj-Cedria Technopark, University of Carthage, Soliman 8020, Tunisia; baha.chamam@gmail.com (B.C.); ismail.trabelsi@certe.rnrt.tn (I.T.); 2Faculty of Sciences of Bizerte, University of Carthage, Jarzouna 7021, Tunisia; 3Laboratory of Application of Materials in Water, Environment, and Energy (LR21 ES15), Faculty of Sciences of Gafsa, University of Gafsa, Gafsa 2112, Tunisia; jrabaraoof@gmail.com; 4Institut Européen des Membranes, IEM, UMR 5635, ENSCM, CNRS, Montpellier University, 34090 Montpellier, France; jean-pierre.mericq@umontpellier.fr (J.P.M.); catherine.faur@umontpellier.fr (C.F.); 5Laboratory of Physics of Materials and Nanomaterials Applied at Environment, Faculty of Sciences of Gabes, Gabes University, Gabes 6072, Tunisia

**Keywords:** composite membrane, PVDF membrane, Ag-ZnO nanoparticles, membrane characterization, permeability, antibacterial property

## Abstract

Ultrafiltration is essential for wastewater treatment, but it faces challenges such as selectivity, control, and fouling reduction. Incorporating nanoparticles into membranes enhances retention, boosts permeability, and limits fouling, improving overall performance. This study explores the properties of PVDF/Ag-ZnO composite membranes, highlighting the influence of silver-doped zinc oxide nanoparticles on membrane structure, performance, and antimicrobial effect. The non-solvent-induced phase separation (NIPS) method successfully led to the preparation of composite membranes; this method used different doses of silver-doped zinc oxide (Ag-ZnO) nanoparticles with Poly(vinylidene fluoride) (PVDF). Scanning electron microscopy (SEM), energy-dispersive X-ray spectroscopy (EDX), and water contact angle measurements were used to validate the influence of nanoparticles on the composite membrane (PVDF/Ag-ZnO) structure. Conversely, morphology (porosity, surface rigorosity), hydrophilicity, and permeability were analyzed through contact angle, image analysis, and flux measurement. In addition, the membranes were tested for antimicrobial activity against *E. coli*. Membrane performance shows that the incorporation of 20% *w*/*w* Ag-ZnO resulted in improved water permeability, which was about 2.73 times higher than that of a pure PVDF membrane (192.2 L·m^−2^·h^−1^·bar^−1^). The membrane porosity showed a linear increase with the number of NPs. The resultant asymmetric membrane was altered to increase the number of pores on the top surface by 61% and the cross-sectional pore surface by 663%. Furthermore, a high antibacterial activity of Ag-ZnO 20% was shown.

## 1. Introduction

Ultrafiltration (UF) is a wide-scale technology used in water treatment applications such as drinking water purification, wastewater treatment, industrial wastewater treatment, seawater desalination pretreatment, and water reuse [1,2]. The advantages of ultrafiltration over other separation techniques like settler or air flotation include a smaller footprint, virus and bacteria retention, and significant reductions in colloids, suspended solids, turbidity, and some levels of total organic carbon [3]. However, one of the main limitations of the ultrafiltration process is still the membrane fouling, which can severely reduce water production and cause significant technical problems such as the need for higher operating pressures and harsh cleaning conditions [4]. Nanotechnology is a field of technology that focuses on synthesizing and applying materials with dimensions of less than 100 nanometers [5]. It has been suggested as a potential solution for addressing various challenges related to water and wastewater treatment [6]. Nanoparticles possess a large specific surface area, small size, high adsorptive capacity, and significant catalytic properties [7,8]. These properties can enhance conventional water treatment processes and enable the development of new high-tech materials. In recent years, the integration of nanotechnology into wastewater treatment processes has been extensively studied by many researchers [9,10]. Novel nanotechnology-based membranes have been developed, and the separation process mechanism has been elucidated. The incorporation of nanoparticles (ZnO, TiO_2_, SiO_2_, CNT) into polyvinylidene fluoride (PVDF) ultrafiltration membranes offers several benefits, including improvements in membrane physico-chemistry and morphological properties, increases in permeability (hydrophilicity) and porosity, and reductions in fouling [7,11,12,13]. In addition to improving membrane quality, coupling nanoparticles with membranes improves their ability to retain contaminants like dyes and heavy metals [14,15,16]. Metals, metal-oxides, nanofibers, and carbon-based materials like graphene and CNTs are used with polymerics to decrease fouling and control microbial growth on the membrane [17]. Various techniques were performed to integrate the NPs into the membrane matrix [18], including in situ polymerization, layer-by-layer (LbL) assembly, electrospinning, and phase inversion, which is one of the most widely used techniques for producing nanoparticle-embedded polymers [19]. For this objective, nanoparticles are mixed with a polymer solution and the mixture is cast into a film. These techniques enable membranes to acquire enhanced functionalities such as improved chemical resistance, antimicrobial activity, and higher filtration efficiency, making them suitable for applications in water treatment. PVDF is one of the most widely used membrane materials due to its excellent chemical and mechanical resistance, as well as its biocompatibility. These advantages make it a preferred choice for membrane fabrication. However, PVDF membranes have relatively low hydrophilicity, which contributes to fouling by proteins and organic matter during wastewater treatment. Rather than replacing the polymer matrix, enhancing PVDF membranes with nanoparticles offers a promising strategy to overcome these limitations while preserving their inherent benefits [20]. Using the phase inversion (PI) approach, [21] prepared a PVDF-SiO_2_ composite membrane and demonstrated that adding SiO_2_ increases the membrane’s resistance to fouling. Ref. [22] showed the photo-filtration properties of PVDF-TiO_2_ membranes. Ref. [23] synthesized modified PVDF/Al_2_O_3_ membranes via the PI method. By analyzing AFM and SEM, it was shown that Al_2_O_3_ nanoparticles increased the anti-fouling and permeation flux of the membrane over the unmodified membrane. Membrane fabrication has proven successful in eliminating dyes with the help of nanoparticles like titanium dioxide (TiO_2_), ferro-ferric oxide (Fe_3_O_4_), carbon nanotubes (CNTs), and graphene oxide (GO) [8,21,23,24,25,26]. Silver-doped zinc oxide (Ag-ZnO) has attracted considerable interest in water and wastewater treatment due to its exceptional properties, including a high specific surface area, strong pollutant adsorption capacity, and photocatalytic ability to degrade organic contaminants under UV light [27]. Incorporating Ag-ZnO nanoparticles (Ag-ZnO NPs) into the membrane polymer matrix not only enhances mechanical strength, durability, and hydrophilicity but also leverages the potent antibacterial properties of silver, making the membrane more resistant to microbial contamination. Hydrophilic membranes tend to have better water permeability, are less prone to fouling by hydrophobic substances, and can reduce the presence of reactive oxygen species (ROS) [28,29,30,31]. This study investigated the effect of incorporating different concentrations of Ag-ZnO NPs on the structure, morphology, and surface composition of PVDF membranes. The filtration performances were measured, and the antibacterial effect against *E. coli* was tested.

## 2. Materials and Methods

### 2.1. Chemicals Reagents

Zinc acetate dihydrate (Zn(CH_3_COO)_2_∙2H_2_O, analytical grade ≥ 99.99%) and silver acetate (AgC_2_H_3_O_2_, 99.99%) were obtained from Sigma-Aldrich (St. Louis, MO, USA). Methanol (CH_3_OH, analytical grade, 99.99%), ethanol (C_2_H_5_OH, analytical grade, 99.999%) and 2-propanol ((CH_3_)_2_CHOH, HPLC grade 99.999%) were obtained from Scharlau (Barcelona, Spain). Poly(vinylidene fluoride) (PVDF) ((CH_2_CF_2_)_n_, molecular weight = 275,000 g/mol) was purchased from Sigma-Aldrich and N, N–dimethylacetamide (DMAc, CH_3_CON(CH_3_)_2_, assay > 99.5%) polyethylene glycol (H(OCH_2_CH_2_)_n_OH, average MW~ 200 g/mol) was obtained from Merck (Boston, MA, USA).

The textile dye reactive black 5 (RB5) was used in an aqueous solution as an azo anionic pollution source and was obtained from Sigma-Aldrich (C_26_H_21_N_5_Na_4_O_19_S_6_, dye content ≥ 50%, MW = 991.8 g·mol^−1^). Dye concentrations are measured with a UV–visible spectrophotometer (Shimadzu, Duisburg, Germany, UV 2401PC) at λ= 597 nm.

### 2.2. Ag-ZnO Nanoparticles Preparation

Silver-doped ZnO aerogel nanoparticles (Ag-ZnO NPs) were prepared by the modified sol–gel method, using zinc acetate di-hydrate [Zn(CH_3_COO)_2_∙2H_2_O] as a precursor and silver acetate [AgC_2_H_3_O_2_] as a doping precursor. Two grams of zinc acetate dihydrate and an adequate quantity of silver acetate [AgC_2_H_3_O_2_] corresponding to the Ag atomic ratio of 0.03 were dissolved into 14 mL of methanol. After that, the solution was quickly dried to make aerogel nanopowder under supercritical conditions of ethanol, with a heating rate of 45 °C/h. Silver-doped ZnO nanoparticles have been characterized previously, and the results are reported by [20].

### 2.3. Membrane Synthesis by Non-Solvent-Induced Phase Separation Technique (NIPS)

The schematic preparation of PVDF and PVDF/Ag-ZnO composite membranes by the phase inversion process (wet non-solvent induced phase separation or wet NIPS) is shown in Figure 1. In the preparation of the NP solution, Ag-ZnO NPs were first dispersed in DMAc solvent in an ultrasonic bath at 20 °C for 20 min to ensure effective uniformity in the solution. PVDF polymer (15 wt.%) and PEG200 additive (5 wt.%) were then added and the dope solution was subjected to further ultrasonic treatment at 20 °C for 20 min and then agitated at a temperature of 50 °C ± 2 °C and a speed of 100 rpm for 24 h. The viscous liquid solution was cast on a Teflon sheet covering a glass plate. This was performed using an automatic casting knife (Erichsen, Hemer, Germany) with a thickness of 250 μm at 25 °C and 4.7 mm/s velocity. The resulting material was immediately immersed in a deionized water bath at 25 °C for 4 h. The prepared membranes were peeled off from the Teflon support and washed thoroughly with deionized water to remove residual solvent. All membranes were conserved in a deionized water solution at room temperature 25 °C ± 2 °C.

The amount of Ag-ZnO NPs varied between 0 and 20 wt.% of PVDF and was calculated using Equation (1) according to polymer usage.(1)$$m_{NPS}=\frac{f\times m_{PVDF}}{100}$$
where f is the Ag-ZnO NPs ratio and m_NPs_ and m_PVDF_ are, respectively, the masses of Ag-ZnO NPs and PVDF.

### 2.4. Membrane Characterization

Scanning electron microscopy (SEM) was used to observe the morphology of the membranes (top surface, bottom surface, and cross-section) and samples were coated with a thin layer of sputtered gold. For cross-section observation, samples were first fractured in liquid nitrogen. Pore counting and surface area calculations were performed using ImageJ 1.52 software. The composition of the membrane was analyzed by EDX (EVO HD15, Zeiss, Oberkochen, Germany).

The measurement of the water liquid drop in an air contact angle (CA) was used to assess the hydrophilicity of membranes. Using an automated liquid pumping system and a Theta optical tensiometer (Attension theta T200, Biolin Scientific, Wetzlar, Germany), the CA of the PVDF/Ag-ZnO membranes was ascertained. Using a microliter syringe, the membrane sample was exposed to a droplet of distilled water, about 5 μL, at room temperature. The sample was positioned under a light source, and a camera and computer software were used to determine the CA. The average of the membrane sample’s five locations was used to obtain the CA value.

Membrane porosity was measured by the gravimetric technique. A 2 cm^2^ piece of membrane was weighed to determine its dry mass and then immersed in 2-isopropanol for 24 h. After removing the sample from the bath, excess surface alcohol was wiped off, and the sample was immediately weighed to determine its wet mass. The membrane porosity was calculated using equation (Equation (2)).(2)$$ɛ=\frac{\left(m_{w}-m_{d}\right)\times \rho_{poly}\;}{\rho_{poly}\times m_{h}+\left(\rho_{IPA}-\rho_{poly}\;\right)\times m_{d}}$$
where ɛ is the membrane porosity (expressed as a percentage), m_w_ is the mass of the wet membranes, m_d_ is the mass of the dry membranes, ρ_IPA_ is the 2-isopropanol density, and ρ_poly_ is the polymer (PVDF) density.

FTIR spectra of dry PVDF and PVDF/Ag-ZnO composite membrane were collected using a diamond ATR-FTIR (Nexus, ThermoFisher, Bend, OR, USA) in the range of 500–4000 cm^−1^ with a resolution of 1 cm^−1^.

A dead-end ultrafiltration (UF) system was used to measure the membrane’s water flux and permeability with deionized water. The system consisted of a 50 mL filter cell (Amicon8400, Millipore, Darmstadt, Germany) connected to an 800 mL feed tank (Millipore, Darmstadt, Germany) at 25 ± 2 °C. Membranes were first conditioned at 0.2 bar, 0.5 bar and 0.7 bar for 10 min, 15 min and 20 min, and lastly conditioned at 1.2 bar for 30 min. Experiments were performed with a 15.89 cm^2^ membrane surface and transmembrane pressures in the range of 0–1 bar. Flux was calculated by measuring the permeate mass using an electronic balance. The flux was determined using the following equation:(3)$$J_{w}=\frac{V}{A\;t}$$
where J_w_ refers to the pure water flux (L·m^−2^·h^−1^), V refers to the total volume of permeate (L), A is the effective membrane filtration area (m^2^), and t refers to the permeation time (2 h).

### 2.5. Membranes Performance

The performance of the fabricated membranes was tested for textile dye removal from highly concentrated aqueous solutions. Filtration–adsorption experiments were performed on a synthetic effluent containing RB5 dye at a 30 mg·L^−1^ initial concentration, pH = 6.3, and 25 ± 2 °C. The filtration tests were performed on a composite PVDF/Ag-ZnO membrane with an effective area of 15.89 cm^2^ through an Amicon8400 filtration cell in dead-end filtration at 1 bar pressure. The membrane’s dye rejection was studied by measuring the absorbance using the UV-Vis spectrophotometer at λmax = 597 nm. The color removal ratio was calculated using the following formula:(4)$$Color\;Removal\;(\%)=\frac{{Abs}_{i}-{Abs}_{f}}{{Abs}_{i}}\times 100$$
where Abs_i_ is the absorbance of the as-prepared RB5 solution and Abs_f_ is the RB5 solution absorbance after filtration.

To decouple the sorption and filtration mechanisms, static sorption studies were carried out on 2 cm^2^ of PVDF/Ag-ZnO membranes weighing 0.028 g. The membrane sample was swirled in a 10 mL RB5 solution with an initial concentration of 30 mg/L for 24 h at pH of 6.2 and 25 ± 2 °C. A mass balance was used to compute the equilibrium sorption capacity using the following equation:(5)$$Q_{e}=\frac{\left(Cₒ-Cₑ\right)\times V}{m}$$
where Q_e_ is the adsorbed quantity at equilibrium (mg/g), C_o_ is the RB5 initial concentration, C_e_ referees the equilibrium RB5 concentration (mg/L), V denotes the solution volume (L), and m is the weight of the PVDF/Ag-ZnO composite membrane (g).

The antibacterial removal efficiency of the membranes was investigated using *Escherichia coli* (*E. coli*) as the test sample. A suspension of *E. coli* (10^9^ CFU/mL) was cultured in 48 mL of Lysogeny Broth (LB) nutrient medium at 30 °C with agitation at 120 rpm in a 250 mL Erlenmeyer flask and incubated overnight. The bacterial concentration was determined by measuring the optical density (OD) at 600 nm. Subsequently, the suspension was diluted 10-fold in a nutrient-free medium (PBS) to yield a final concentration of 10^2^ CFU/mL in 20 mL. Negative and positive tests controls were carried out without and with 200 µL of 10^2^ CFU/mL bacteria in tow Petri dishes containing LB agar and incubated overnight at 30 °C.

Following this, a membrane piece measuring 1.5 cm × 1.5 cm was placed at the center of the prepared solution in one Petri dish, while powdered nanoparticles, initially mixed with the solution, were spread onto another Petri dish in order to investigate their antibacterial effect. A control experiment was conducted using 11 mL of soft agar prepared in a sterile glass tube with 1 mL of sterile water. All dishes were then incubated at 30 °C for 24 h.

## 3. Results

### 3.1. PVDF and PVDF/Ag-ZnO Composite Membranes Composition

UF membranes were prepared by a wet non-solvent-induced phase separation process. For the preparation of PVDF/Ag-ZnO nanocomposite membranes, different amounts of Ag-ZnO nanoparticles (5, 10, and 20 wt.% NPs/PVDF) were added to the original PVDF membrane composition. The EDX analysis shown in Figure 2 confirms the purity of the PVDF membrane; the chemical composition only includes F, C, and O with an atomic ratio of fluorine and carbon very close to the theoretical ones (%F = 50% and %C = 50%; hydrogen is uncountable for EDX analysis). For PVDF/Ag-ZnO membranes, EDX spectra confirm the presence of Ag-ZnO nanoparticles in the final membrane composition. The ultrasonic treatment of the casting solution induces the homogenization and dispersion of NPs in the collodion. The F, C, and Zn elements are detected for all PVDF/Ag-ZnO membranes. Based on the percentage of F in PVDF (50% atomic ratio) and Zn in ZnO (50% atomic ratio), the membrane composition can be calculated from the EDX measurement in Table 1.

The ratio of NPs to PVDF is consistent with the conditions of membrane preparation. For PVDF/Ag-ZnO 5% and PVDF/Ag-ZnO 10% membranes, the added quantity of Ag-ZnO gives a final quantity of Ag equal to 0.015 and 0.03%, respectively, for Ag-ZnO 5% and Ag-ZnO 10%. These quantities are under the detection limit of the instrument used. In the corollary, the element Ag could not be detected in these samples. For the membrane elaborated with 20% Ag-ZnO NPs, the EDX confirms the presence of silver in the final composition; this is due to the higher amount of silver in the final product (the theoretical amount of silver is 0.06%).

On the other hand, EDX analysis provides atomic percentages very close to the theoretical values. This confirms the homogeneous distribution of nanoparticles in the developed membranes, indicating that there is no exclusion of NPs towards the surface or sedimentation towards the bottom.

The recorded FTIR spectra of pure PVDF and PVDF/Ag-ZnO composite membranes, presented in Figure 3, show the presence of PVDF characteristic bands at 763, 838, 875, 1072, 1182, 1401, 2852, and 3023 cm^−1^. The asymmetric and symmetric vibrations of PVDF concerning CH_2_ are shown, respectively, by the weak bands at 3023 and 2852 cm^−1^. The CH_2_ restless vibration was identified by the absorption band at 1401 cm^−1^. The band observed at 1182 cm^−1^ can be attributed to the C–C bond of PVDF. The observed bands at 875 and 838 cm^−1^ can be attributed, respectively, to the C-C-C asymmetric stretching vibration and C-F stretching vibration of PVDF [32]. The absorption peak at 1072 cm^−1^ is also attributed to the C–O group. The observed peak at 763 can be assigned to the α-phase of the PVDF polymer [33]. In the PVDF/Ag-ZnO membrane’s spectra, the peaks formed are located in almost the same position as bare PVDF peaks, indicating that Ag-ZnO NPs components may be physically mixed. This can be explained by the fact that the oxide bands only appear in the far-infrared region [34]. Also, this phenomenon can be explained by the trace amount of Ag-ZnO NPs in the membrane matrix [18,35,36].

### 3.2. PVDF and PVDF/Ag-ZnO Composite Membrane Characterization

Microscopic analysis was performed to determine the morphological structure of the prepared membranes. Figure 4 shows microscopic images of the top surface (a1–a4), bottom surface (b1–b4), and cross-section (c1–c4) of the membranes.

These images show a good distribution of pores on the upper and lower surfaces. The number of pores, counted over a reference surface of 50 μm^2^, increased from 300 to 483, representing a 61% increase on the upper surface for PVDF and PVDF/Ag-ZnO 20% (Table 2), respectively, and the trend was the same on the bottom surface. This led to a rise in pore density from 6 to 9.7 pores/μm^2^. This can be explained by the presence of nanoparticle sites on the interface, which attract greater water influx due to their hydrophilic properties. Then, the cross-section SEM images show that all membranes share the common asymmetric structure of membranes obtained by wet NIPS with three layers: a top thin “dense” and selective layer, a finger-shaped macrovoid structure, and a macroporous sponge-like structure below. The increase in the Ag-ZnO nanoparticle ratio in the membrane seems to increase macrovoid sizes (as seen on the cross-section) and also sponge-like structural macropores (as seen on the bottom surface). In fact, the higher surface pore density of membranes with Ag-ZnO NPs, compared with those without NPs, is directly attributed to the presence of nanoparticle sites on the interface, which increase water influx due to their hydrophilic properties during the membrane structuring phase (solvent evaporation).

Incorporating 5% and 10% Ag-ZnO NPs slightly enlarged the macrovoids. The average length of the macrovoids increased from 34 µm to 54.6 µm, while the average width rose from 9 µm to 10 µm. Additionally, the average surface area expanded significantly from 143 µm^2^ to 173 µm^2^ (a 21% increase) when comparing PVDF membranes to 20% PVDF/ZnO. However, a more substantial increase in both the number and size of macropores was observed in the 20% Ag-ZnO composite membrane.

Porosity measurements carried out using isopropanol confirm that all membranes are highly porous (Table 2). Porosity values are found to be increasing from 80.8 ± 0.21 for pure PVDF membranes to 91 ± 0.24 for PVDF/Ag-ZnO 20%. This is in keeping with the morphological results obtained using SEM and the top surface porosity calculated. As the mass of NPs in the membrane composition is raised, the presence of pores is favored, which increases the porosity of the membranes. Nevertheless, this porosity depends mainly on cross-sectional morphology and not on changes in the top and bottom surface. In fact, porosity remains mainly a function of the size and volume of the macropores and micropores of the material.

The water contact angle is used to assess the hydrophilicity of the membrane surface, with smaller contact angles indicating higher hydrophilicity. The measured contact angle values were on average 77.12 ± 0.54° for pure PVDF, 77.4 ± 0.06° for PVDF/Ag-ZnO 5%, 75.29 ± 0.85° for PVDF/Ag-ZnO 10%, and 68.61 ± 2.13° for PVDF/Ag-ZnO 20%. The 20% Ag-ZnO nanoparticles addition improved the hydrophilicity of the specimen, as evidenced by a slight decrease in the contact angle compared to the PVDF membrane (Table 2). However, the addition of 5% and 10% Ag-ZnO nanoparticles seemed to have no effect. The incorporation of Ag-ZnO into PES membranes is used to enhanced their hydrophilic properties. Displaying similar results, [37] showed that the hydrophobicity of PVDF/Ag decreased with increasing amounts of Ag NPs in the membrane. The hydrophilic nature of Ag-ZnO nanoparticles was mainly responsible for the improved hydrophilicity of the PVDF membrane, but it seems that a certain number of nanoparticles (more than 20 wt.%) are required to have a significant effect on the hydrophilicity increase. The hydrophilic nature of a material is a crucial parameter, as hydrophilicity can both increase water passage through the membrane and decrease fouling since hydrophobic materials are more susceptible to fouling when hydrophobic molecules or particles (proteins, colloids, etc.) are present in the fluid being filtered.

### 3.3. Membranes Permeability

The results (Figure 5) show that permeability reaches 192.2 L·m^−2^·h^−1^·bar^−1^ for the PVDF/Ag-ZnO 20% membrane. This value is about 2.7 times higher than that of the pure PVDF membrane, the value of which is initially equal to 75.4 L·m^−2^·h^−1^·bar^−1^.

It is interesting to note that the little increase in permeability was observed for 5% and 10% of nanoparticles loaded to the PVDF membrane, with values of, respectively, 83.65 L·m^−2^·h^−1^·bar^−1^ and 102.41 L·m^−2^·h^−1^·bar^−1^. As confirmed by SEM and porosity analysis, the permeability increase observed for low NP concentrations can be attributed to an increase in pore size, surface porosity, and volume porosity, as illustrated in the membrane characterization (Table 2). The parameter that seems to have the greatest influence on permeability is porosity, since there is a linear relationship between porosity and permeability. For high NP concentrations, the permeability increase is further amplified by the increase in membrane hydrophilicity. This finding is consistent with the results reported by [38], who investigated the use of silver nanoparticles in pure PVDF membranes for water desalination applications.

### 3.4. Adsorption Study of RB5 on Composite Membranes in Static Mode

Figure 6 shows that the static equilibrium sorption capacities of the RB5 dye by the membranes are 2.13 mg·g^−1^ for virgin PVDF and 3.44 mg·g^−1^, 5.07 mg·g^−1^, and 6.40 mg·g^−1^ for PVDF/Ag-ZnO 5%, PVDF/Ag-ZnO 10% and PVDF/Ag-ZnO 20%, respectively. The mass of nanoparticles added to the initial composition gradually enhances the sorption capacity of the composite membranes.

The adsorption of the RB5 dye by Ag-ZnO NPs was calculated from an isotherm modulation study conducted on the NPs and RB5 dye [2]. This study demonstrated that the adsorption mechanism is described by a Langmuir isotherm, with a maximal adsorption capacity of 23.8 mgRB5/gAg-ZnO.

The treatment of adsorption isotherms for composite membranes composed of PVDF and nanoparticles can be approached in two different ways. The first method (Figure 6 left) is based on the assumption that the adsorption capacity of the nanoparticles remains constant at 23.8 mg per gram of nanoparticles. Given this fixed value and according to the NP content, NP can be fixed at 0.0309, 0.0619 and 0.1190 mgRB5 for membranes containing 5%, 10%, and 20% NPs, respectively. It is now possible to determine the adsorption capacity of the PVDF matrix by subtracting the contribution of the nanoparticles from the total adsorption measured. This approach provides insight into the specific role of the polymer in the overall adsorption performance of the membrane, where the resulting adsorption capacities for the PVDF matrix are 2.13 mg·g^−1^, 2.25 mg·g^−1^, 2.69 mg·g^−1^, and 1.64 mg·g^−1^ for PVDF/Ag-ZnO membranes containing 5%, 10%, and 20% NPs, respectively.

Conversely, an alternative method focuses on calculating the mass of the adsorbed dye attributed to PVDF and, more importantly, to the surface area developed by the polymer. The specific surface area of PVDF is proportional to the membrane’s permeability, meaning that variations in permeability can directly influence adsorption performance. By estimating the adsorption contribution of PVDF in this manner (Figure 6 right), the mass adsorbed by the nanoparticles can then be deduced, allowing for a more detailed understanding of the interactions between the membrane components and the adsorbed molecules (24.1 mgRB5/gNP, 24.7 mgRB5/gNP, 10.3 mgRB5/gNP, for membranes containing 5%, 10%, and 20% NPs, respectively). The results confirm that RB5 demonstrates access to nanoparticles and adsorption proportional to permeability for Ag-ZnO 5% and 10% membranes, but not for Ag-ZnO 20%. The isotherms clearly indicate a structural change in the membrane but confirm that nanoparticles are less accessible. In fact, both approaches provide complementary insights into the adsorption behavior of composite membranes. These results highlight that the increased sorption capacity is primarily driven by the change in internal porosity induced by the addition of Ag-ZnO nanoparticles.

The incorporation of Ag-ZnO nanoparticles results in a strong interaction between the positively charged Ag^+^ and Zn^2+^ ions and the negatively charged -SO_4_^2−^ anionic dye molecule, potentially increasing the adsorption capacity. Such results are supported by previous findings. Ref. [37] demonstrated that incorporating Fe_2_O_3_:Ag(S) NPs into the PVDF membrane composition increased the adsorption capacities of various dyes, including Malachite Green (from 7 mg·g^−1^ to 10 mg·g^−1^), Methyl Violet (from 3 mg·g^−1^ to 8.8 mg·g^−1^), and Pyronin Y (from 3.12 mg·g^−1^ to 8.9 mg·g^−1^).

To evaluate the improvement in membrane retention capacity, filtration tests were conducted on 50 mL of a RB5 textile dye solution (30 mg·L^−1^) at pH 6.3, temperature of 25 °C, and pressure of 1 bar for a duration of 1 h. The dye removal capacities, calculated as the percentage difference between the initial and final dye concentrations relative to the initial concentration, were 24.4% for pure PVDF, 31.5% for PVDF/Ag-ZnO 5%, 39.8% for PVDF/Ag-ZnO 10%, and 56% for PVDF/Ag-ZnO 20% (Table 3). The enhanced dye retention was attributed to two main factors: the molecular cut-off of the PVDF/Ag-ZnO ultrafiltration membranes, which retained the RB5 dye molecules, and the dye adsorption capacity of the membranes, as confirmed by static sorption tests.

### 3.5. Antibacterial Test

The antibacterial test included control tests to confirm the reliability of the nutrient medium, and antibacterial tests with Ag-ZnO NPs with a Ag atomic ratio of 0.03 and PVDF/Ag-ZnO 20%. The negative and positive control test confirmed that the nutrient medium itself does not lead to contamination, ensuring the experiment’s reliability, whereas the nutrient medium shows the bacteria’s natural growth without any antibacterial agent interference. Then, antibacterial tests showed that Ag-ZnO NPs alone strongly inhibit bacterial growth, with no *E. coli* strains detected (Figure 7). For membranes, pure PVDF exhibited no antibacterial activity (41 colonies), while PVDF/Ag-ZnO 20% membranes displayed a significant lysis zone (no colonies detected within a six-millimeter radius around the membrane) and reduced colony count (13), indicating good antibacterial efficacy.

The antibacterial effect of Ag-ZnO NPs is attributed to the presence of Zn⁺ and Ag⁺ cations, which disrupt bacterial cell walls, encapsulate bacteria, and oxidize key cellular components [38]. The addition of these nanofillers transforms PVDF’s behavior, overcoming its natural hydrophobicity, which promotes bacterial adhesion. Ag-ZnO NPs effectively prevent bacterial deposition and reduce biofouling through antimicrobial nanofillers. This modification imparts antibacterial properties absent in pure PVDF [39].

This highlights the potential of Ag-ZnO-enhanced membranes for addressing biofouling challenges.

## 4. Conclusions

This study successfully developed PVDF/Ag-ZnO ultrafiltration membranes incorporating Ag-ZnO nanoparticles (5%, 10%, and 20%) via the non-solvent-induced phase separation method. Chemical analysis confirmed that Ag-ZnO nanoparticles were well-dispersed and uniformly distributed within the PVDF matrix. All membranes exhibited an asymmetric morphology characterized by three distinct layers. The incorporation of Ag-ZnO nanoparticles notably altered the membrane structure, resulting in higher porosity, longer and wider pores, and larger macrovoids. These changes are directly linked to the hydrophilic nature of the nanoparticles, which enhance water influx during the membrane formation phase. Consequently, membrane permeability improved, with the PVDF/Ag-ZnO-20% membrane achieving a flux of 192.2 L·m^−2^·h^−1^·bar^−1^, about 2.7 times higher than that of pure PVDF (70.4 L·m^−2^·h^−1^·bar^−1^). The influence of NPs on permeability shows that permeability increases linearly with increasing porosity, but increases significantly when hydrophilicity also increases. The Ag-ZnO addition also enhanced membrane performance for anionic azo dye RB5 retention, with the PVDF/Ag-ZnO 20% membrane showing a retention rate of 56% versus 24.4% for pure PVDF. Furthermore, the incorporation of Ag-Zn nanoparticles imparted antibacterial attributes to the membranes, substantially enhancing the resistance against biofouling by bacterial growth. These innovative membranes offer a game-changing solution for the textile industry, providing efficient pretreatment for reverse osmosis.

## Figures and Tables

**Figure 1 polymers-17-01269-f001:**
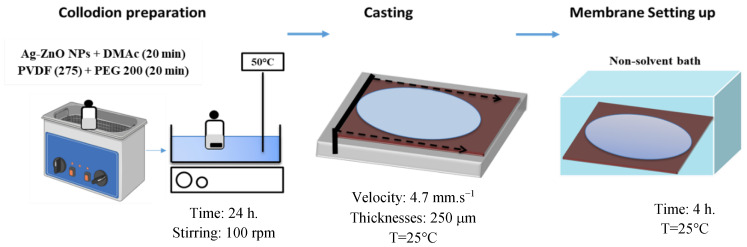
Preparation process of PVDF/Ag-ZnO membranes.

**Figure 2 polymers-17-01269-f002:**
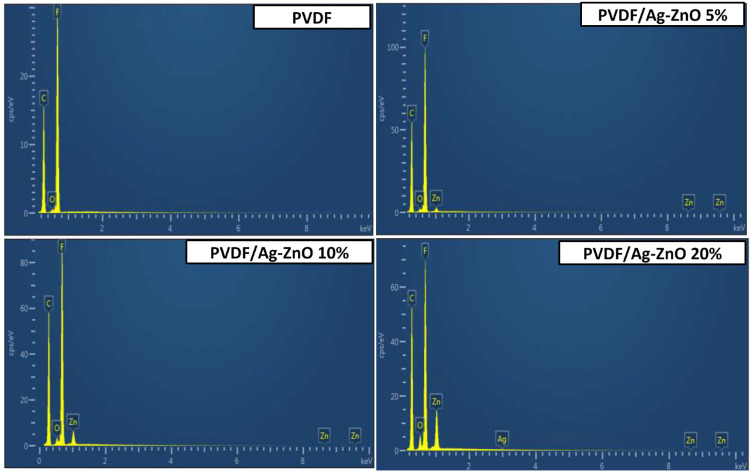
EDX spectra of PVDF, PVDF/Ag-ZnO-5%, PVDF/Ag-ZnO-10%, and PVDF/Ag-ZnO-20% membranes.

**Figure 3 polymers-17-01269-f003:**
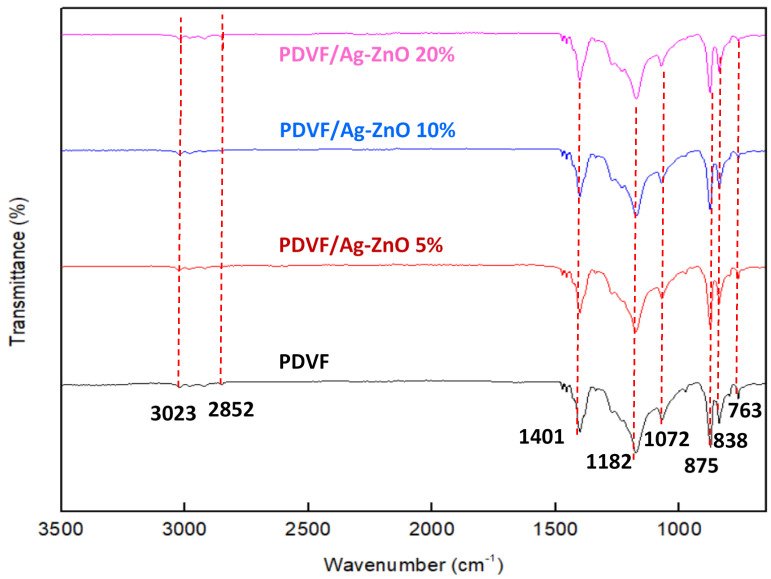
ATR-FTIR analysis of PVDF, PVDF/Ag-ZnO-5%, PVDF/Ag-ZnO-10%, and PVDF@/Ag-ZnO-20% membranes.

**Figure 4 polymers-17-01269-f004:**
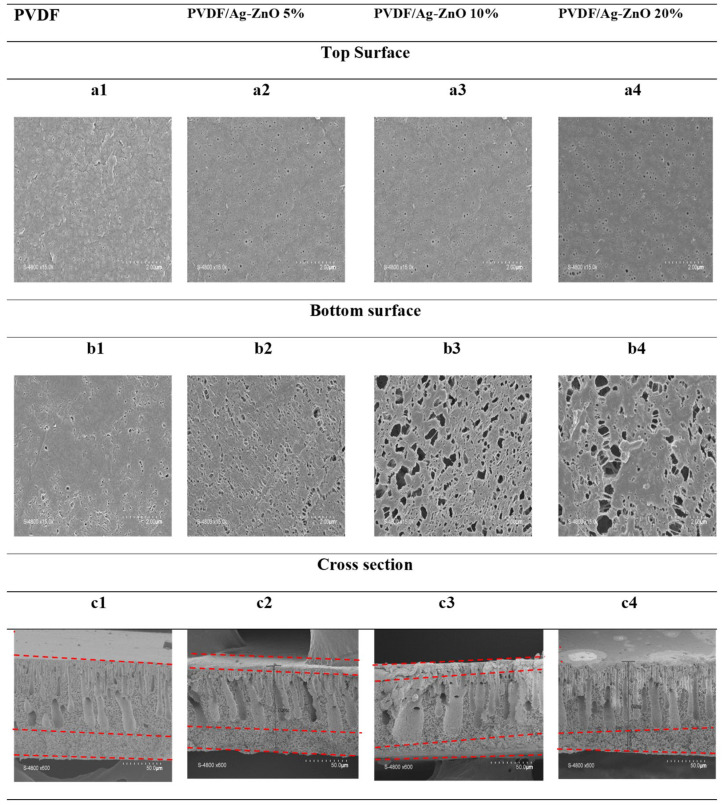
Top and bottom surface and cross-section SEM figure of PVDF, PVDF/Ag-ZnO 5%, PVDF/Ag-ZnO 10%, and PVDF/Ag-ZnO 20% membranes.

**Figure 5 polymers-17-01269-f005:**
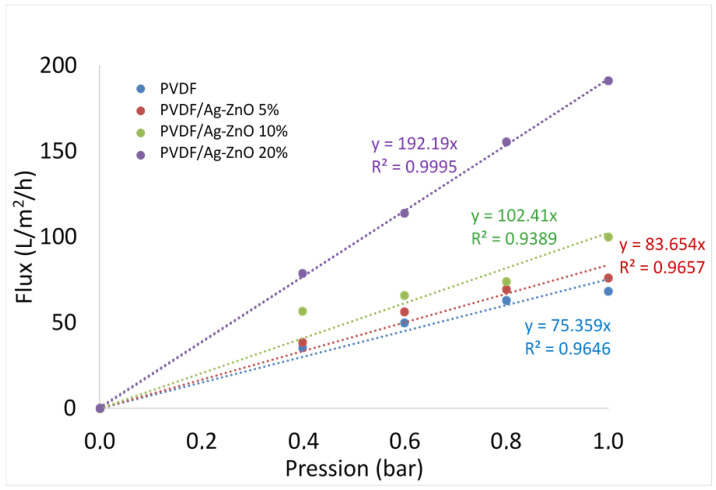
The evolution of the flow as a function of pressure for PVDF, PVDF/Ag-ZnO 5%, PVDF/Ag-ZnO 10%, and PVDF/Ag-ZnO 20% membranes.

**Figure 6 polymers-17-01269-f006:**
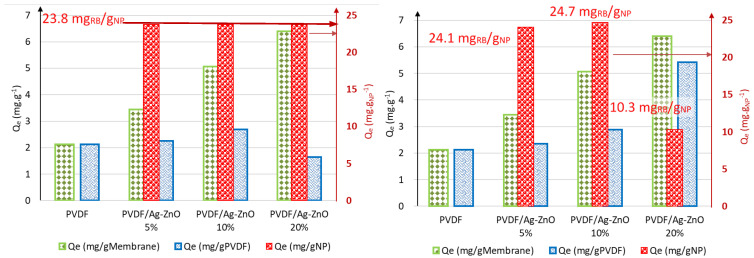
RB5 adsorbed quantity of by pure PVDF and PVDF/Ag-ZnO composite membranes.

**Figure 7 polymers-17-01269-f007:**
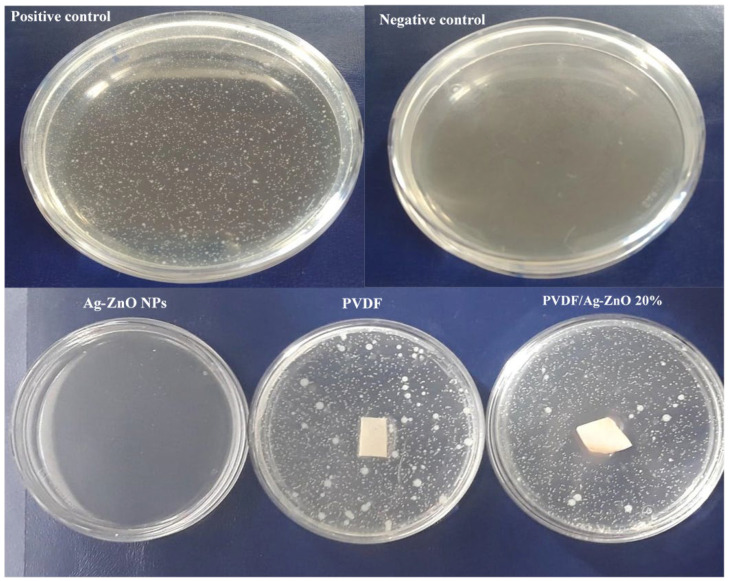
Antibacterial test of Ag-ZnO NPs, PVDF, and PVDF/Ag-ZnO-20% composite membranes.

**Table 1 polymers-17-01269-t001:** EDX chemical composition of PVDF, PVDF/Ag-ZnO-5%, PVDF/Ag-ZnO-10%, and PVDF/Ag-ZnO-20%.

Membranes	PVDF		PVDF/Ag-ZnO-5%		PVDF/Ag-ZnO-10%		PVDF/Ag-ZnO-20%	
	%Mass	%Atomic	%Mass	%Atomic	%Mass	%Atomic	%Mass	%Atomic
F	57.99	46.71	56.15	46.06	49.63	41.19	40.90	35.86
Zn	0.00	0.00	2.52	0.60	6.35	1.53	14.43	3.68
Ag	0.00	0.00	ND *	ND *	ND *	ND *	0.29	0.05

ND *: not detectable.

**Table 2 polymers-17-01269-t002:** Membrane characterization.

		PVDF	PVDF/Ag-ZnO5%	PVDF/Ag-ZnO10%	PVDF/Ag-ZnO20%
Pore surface in cross-section µm^2^/50 µm^2^		147	162	155	173
Average	Width µm	9	9.3	9.6	10
	Lenghth µm	34	42.2	48.4	54.6
Top surface	Pore count	300	386	420	483
	Increase (%)		29	40	61
Bottom surface	Pore count	260	280	323	363
	Increase (%)		8	24	40
Pore surface in bottom µm^2^/50 µm^2^		0.008	0.0165	0.0313	0.061
Porosity (%)		80.8 ± 0.21	82 ± 2	86 ± 1.8	91 ± 0.24
Porosity increase (%)			1.48	6.4	12.6
Contact angle		77.12 ± 0.54°	77.4 ± 0.06	75.29 ± 0.85	68.61 ± 2.13
Permeability (L·h^−1^·m^−2^·bar^−1^)		75.4	83.6	102.4	192.2
Enhanced permeability (%)			11	36	155

**Table 3 polymers-17-01269-t003:** Filtration performance.

Membrane	PVDF	PVDF/Ag-ZnO 5%	PVDF/Ag-ZnO 10%	PVDF/Ag-ZnO 20%
Dye removal (%)	24.2	31.5	39.8	56.0
Efficacity enhancement (%)		30	64	131

## Data Availability

The original contributions presented in this study are included in the article. Further inquiries can be directed to the corresponding author.

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
