# Peer review of "Incorporation of Ag-ZnO Nanoparticles into PVDF Membrane Formulation to Enhance Dye Retention, Permeability, and Antibacterial Properties"

_polymers, 2025, doi:10.3390/polym17091269_

Round 1
Reviewer 1 Report
Comments and Suggestions for Authors
Comments
This paper reports “Incorporation of ZnO: Ag Nanoparticles in PVDF Membrane formulation to enhance dye retention, permeability, and antibacterial properties” There are many shortcomings and loopholes in the logical aspects of material characterization, mechanism explanation, and experimental design. In addition, the manuscript contains several avoidable errors that should be corrected to enhance the quality. Based on the current state of the manuscript, it is not recommended for publication in this journal. Some specific opinions are as follows:
- The introduction should clearly state what distinguishes this research from existing ZnO or TiO₂-doped membranes, highlighting the contribution and value of this study.
- The article mentions the use of ultrasonic dispersion for ZnO:Ag, but relies solely on EDX analysis to confirm the presence of nanoparticles, which lacks sufficient convincing evidence. It is recommended to add TEM or XRD characterization to provide more intuitive information on particle distribution and structure.
- In lines 362-370, the paper mentions the adsorption capacities of ZnO:Ag-doped membranes with different contents: 24.1, 24.7, and 10.3 mgRB5/gNP (for 5%, 10%, and 20% ZnO:Ag content, respectively). However, the adsorption capacity significantly decreases at 20%. Please provide further explanation.
- The study only demonstrates the initial performance of the ZnO:Ag membranes, without considering their stability after prolonged use. It is recommended to add long-term immersion experiments or multiple-cycle tests to evaluate the durability of the membranes.
- The study shows antibacterial properties of ZnO:Ag only through inhibition zone experiments but does not analyze the antibacterial mechanism. Please explain why the addition of ZnO:Ag provides antibacterial properties and further reveal the antibacterial mechanism.
- High nanoparticle content (20%) may affect the mechanical strength of the membrane (such as elongation at break, tensile strength). It is suggested to supplement relevant mechanical performance tests to evaluate practical applicability.
- There is a discrepancy in the contact angle data in Table 2; the contact angle of PVDF/ZnO:Ag 5% membrane (77.4°) is slightly higher than that of pure PVDF (77.1°), which contradicts the trend of improving hydrophilicity. Please check for experimental errors or reproducibility.
- The paper mentions that ZnO:Ag doping reduces the water contact angle of PVDF membranes (improving hydrophilicity), but the change in data is minimal (only from 77.12° to 68.61°), and the contact angle of 5% ZnO:Ag is even higher than pure PVDF (77.4°). Please explain why low doping levels (5%) fail to enhance hydrophilicity.
- The adsorption model in Figure 6 is not sufficiently presented; the current adsorption isotherm only displays experimental data points. It is recommended that the authors include Langmuir or Freundlich fitting curves and report the fitting equations and R² values to verify the adsorption mechanism.
- In the 20% NPs-doped membrane, the porosity only increased by 12.6% (from 80.8% to 91%), but the permeability increased by 155% (from 75.4 L·m⁻²·h⁻¹·bar⁻¹ to 192.2 L·m⁻²·h⁻¹·bar⁻¹). According to the Hagen-Poiseuille equation, flux should be directly proportional to porosity. A reasonable explanation for this nonlinear relationship is required.
- In line 35, there are two consecutive periods ("..") at the end, which should be corrected.
- Tables 2 and 3 are missing some data; please provide the missing information.
- In Figure 5, the comma should be replaced with a period.
- What is the meaning of the arrows in Figure 6? Please clarify.
- The captions in the text refer to two "Figure 6". Please correct this to "Figure 5" and "Figure 6".
- It is suggested to optimize the graphics to improve the readability and scientific quality of the images.
- The reference formatting is inconsistent; it is recommended to adjust according to the journal's requirements.
- There are some minor formatting issues in the text. Please check and correct them.
Reviewer 2 Report
Comments and Suggestions for Authors
ZnO:Ag particles were incorporated into a PVDF phase-inversion membrane by mixing them into the casting solution. The chemical and physical properties of the membrane were characterized, along with its permeation performance and antibacterial properties. However, several concerns need to be addressed as below:
- ZnO:Ag particles were synthesized but not characterized. Characterizations of ZnO:Ag particles are encouraged to be included to demonstrate the successful synthesis and evaluate their particle sizes.
- To be published in current journal, the novelty and impact of current manuscript should be better explained. The effect of ZnO concentration and other particles in phase-inversion PVDF membranes has been reported previously. If silver doping is the novel aspect of this study, a comparison between phase-inversion PVDF/ZnO and PVDF/ZnO:Ag membranes would help clarify the effects of silver doping.
- The authors claim a homogeneous distribution of ZnO:Ag particles based on the EDX spectrum. However, EDX spectra only provide localized chemical composition data. To better support this claim, surface and cross-section EDX mapping images should be included to demonstrate the nanoparticle distribution throughout the membrane.
- The changes in membrane pore size after ZnO:Ag incorporation should be discussed. Additionally, the authors should clarify whether the intended application of this membrane requires selectivity and how increased porosity or pore size due to ZnO:Ag incorporation would impact its performance.
- Table 2 contains several blank entries under the “pore surface in cross-section” section. Filling in these missing values would strengthen the claims made in the paper and provide a more complete dataset.
Round 2
Reviewer 1 Report
Comments and Suggestions for Authors
The paper can be accepted